# Enhancement of homology-directed repair with chromatin donor templates in cells

**Grisel Cruz-Becerra, James T Kadonaga***

Section of Molecular Biology, University of California, San Diego, La Jolla, United States

**Abstract** A key challenge in precise genome editing is the low efficiency of homology-directed repair (HDR). Here we describe a strategy for increasing the efficiency of HDR in cells by using a chromatin donor template instead of a naked DNA donor template. The use of chromatin, which is the natural form of DNA in the nucleus, increases the frequency of HDR-edited clones as well as homozygous editing. In addition, transfection of chromatin results in negligible cytotoxicity. These findings suggest that a chromatin donor template should be useful for a wide range of HDR applications such as the precise insertion or replacement of DNA fragments that contain the coding regions of genes.

## Introduction

The ability to manipulate genomes precisely is revolutionizing the biological sciences (*Doudna, 2020*). Of particular utility is the modification or insertion of customized DNA sequences at a specific genomic location by homology-directed repair (HDR) (*Jasin and Rothstein, 2013*). For genome engineering in cells, HDR typically involves the generation of a specifically targeted DNA double-strand break (DSB) in the presence of a homologous DNA donor template that contains the desired sequence to be modified or inserted (*Urnov et al., 2005*; *Bedell et al., 2012*; *Jinek et al., 2012*; *Cong et al., 2013*; *Pickar-Oliver and Gersbach, 2019*).

A key challenge in successful genome editing has been the low efficiency of HDR (*Carroll, 2014*; *Harrison et al., 2014*). For the generation of specific alterations in a short stretch of DNA (<50 nt), recently developed techniques such as base editing (*Rees and Liu, 2018*; *Molla and Yang, 2019*) and prime editing (*Anzalone et al., 2019*) have been shown to be highly effective. In addition, for the imprecise insertion of larger DNA fragments, homology-independent approaches can be used (*Auer et al., 2014*; *He et al., 2016*; *Suzuki et al., 2016*). These powerful methods cannot, however, be used for the precise insertion or replacement of >50 bp DNA fragments, such as those containing the coding regions of genes. For such applications, we considered a different strategy for increasing the efficiency of HDR in cells. Based on our previous observation that homologous strand pairing, an early step in HDR, occurs more efficiently with a chromatin donor template than with a plain (naked) DNA donor template in vitro (*Alexiadis and Kadonaga, 2002*), we postulated that HDR in cells might similarly be more efficient with a chromatin relative to a naked DNA donor template.

In this study, we tested this idea by comparing the efficiency of HDR with chromatin versus naked DNA donor templates in conjunction with DSBs generated by the clustered regularly interspaced short palindromic repeats (CRISPR)-Cas9 system. We found that the overall HDR efficiency as well as the frequency of homozygous editing is enhanced by the use of a chromatin donor template relative to a DNA donor template. We thus envision that a chromatin donor template, which resembles the natural form of DNA in the nucleus, could be widely used to increase the success of HDR-mediated

***For correspondence:**
jkadonaga@ucsd.edu

**eLife digest** Genome editing is a powerful tool used across a wide range of biomedical research. There are several different techniques used, depending on the type of edit being made, and one known as homology-directed repair – or HDR for short – is a common technique for precisely inserting large sections of DNA, such as those needed to make desired proteins in cells.

HDR takes advantage of the cell's mechanisms for repairing damage to DNA if both strands of the DNA double helix are broken. The mechanism relies on a DNA template to stitch the strands back together. To insert or replace a new DNA sequence, scientists can add a customized piece of DNA of their choosing to the cell so that it might be incorporated into the genome. However, HDR is not very efficient, and the success rate is often less than a few percent.

In HDR gene editing, the DNA template is typically added as purified, or 'naked', DNA. However, the natural form of DNA in cells, known as chromatin, is where the DNA helix is wrapped around a cluster of proteins known as histones. Cruz-Becerra and Kadonaga tested the idea that DNA in the form of chromatin might be more effective as a template for HDR than naked DNA.

The two approaches were compared to see which was better at inserting a sequence at three different locations in the genome of lab-grown human cells. In these experiments, the chromatin templates were 2.3- to 7.4-fold more efficient than the naked DNA. Also, the DNA in human cells is organized as pairs of chromosomes, and chromatin was better than naked DNA for editing both copies of the chromosome pairs rather than only one of them. In addition, the chromatin is potentially less toxic to the cells. Cruz-Becerra and Kadonaga hope that this will be useful for increasing the success rate of HDR experiments and potentially other methods of gene editing in the future.

applications, particularly those that involve the targeted insertion of DNA fragments such as the coding regions of genes.

## Results

To ascertain whether the use of chromatin donor templates affects the efficiency of HDR in cells, we reconstituted three DNA donor templates (corresponding to the human *GAPDH*, *RAB11A*, and *ACTB* loci) into chromatin and tested the relative efficiencies of the targeted insertion of the GFP coding sequence with chromatin versus naked DNA versions of these templates (*Figure 1* and *Figure 1—figure supplements 1–4*). The chromatin was reconstituted by using salt dialysis methodology with plasmid DNA and purified core histones from *Drosophila* embryos, which contain a broad mixture of covalent modifications that have not been precisely resolved (*Levenstein and Kadonaga, 2002*). With standard CRISPR-Cas9 methodology and human MCF10A cells (non-tumorigenic epithelial cells derived from human mammary glands), we observed that the use of a chromatin donor template relative to a naked DNA donor template resulted in a 7.4-, 2.9-, and 2.3-fold increase (average of three biological replicates) in the directed insertion of GFP sequences at the *GAPDH*, *RAB11A*, and *ACTB* loci, respectively (*Figures 1B, C and D* and *Figure 1—figure supplements 3* and *4*). Thus, at three different loci (*GAPDH*, *RAB11A*, and *ACTB*) in human MCF10A cells, there was a higher efficiency of HDR-mediated GFP insertion with chromatin donor templates than with naked DNA donor templates.

For many applications of HDR, it is essential to modify all of the copies of the target gene. Therefore, to test the frequency of occurrence of precise homozygous gene editing in the diploid MCF10A cells, we carried out PCR analyses of the individual GFP-positive clones, and we observed a variable but consistently higher frequency of homozygous HDR insertions with chromatin donor templates than with naked DNA donor templates at all three loci (*GAPDH*, *RAB11A*, and *ACTB*) in MCF10A cells (*Figure 2* and *Figure 2—figure supplements 1–5*). At the *GAPDH* locus, the use of chromatin relative to naked DNA donor templates resulted in a 2.1-fold increase in homozygous editing. At the *RAB11A* locus, there was a high frequency of homozygous insertions with the naked DNA donor template, and the use of a chromatin donor template only slightly augments (1.1-fold increase) the percentage of homozygous clones. Strikingly, at the *ACTB* locus, homozygous insertions were observed only with a chromatin donor template. These findings thus show that the use of

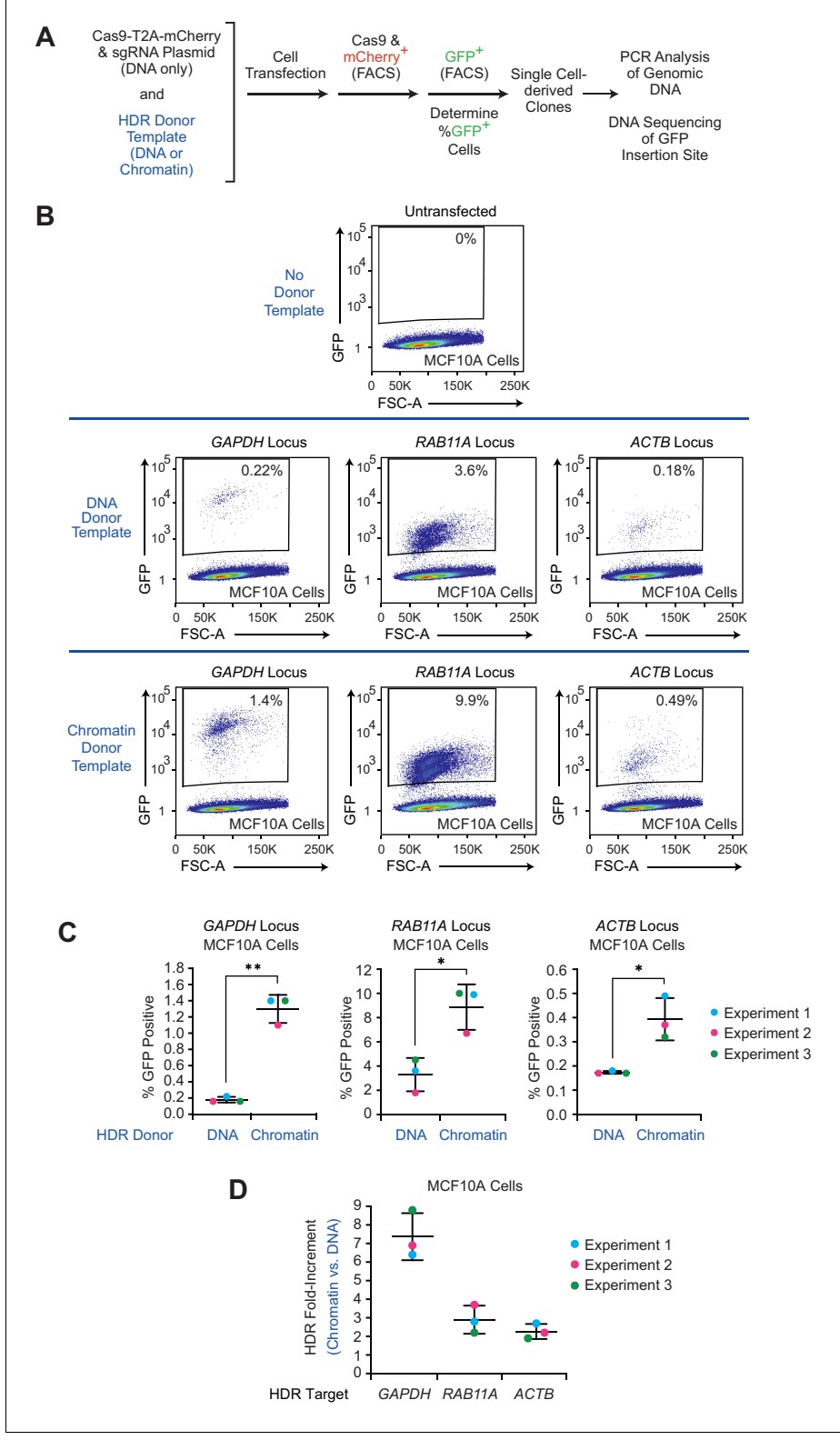

**Figure 1.** The efficiency of HDR-mediated gene editing with CRISPR-Cas9 is higher with chromatin donor templates than with DNA donor templates. (A) Schematic outline of the workflow in the CRISPR-Cas9-mediated editing experiments with DNA or chromatin donor templates. The HDR-mediated insertion of the GFP sequence was directed to different loci as follows. Plasmid DNA containing the coding sequence for Cas9-T2A-mCherry and a target-specific sgRNA sequence was co-transfected into different human cell lines with the corresponding HDR

*Figure 1 continued on next page*

*Figure 1 continued*

donor template as either DNA or chromatin. At 24 hr post-transfection, mCherry-positive cells were enriched by FACS and cultured for an additional 10 days. The expression of GFP was then analyzed by flow cytometry, and individual GFP-positive cells were sorted by FACS to generate independent clones. To determine whether there was partial or complete conversion of the multiple chromosomes containing the target genes, genomic DNA samples from each of several independent GFP-positive clones were analyzed by PCR. In addition, the precise integration of the GFP sequence at the target sites in representative edited clones was confirmed by DNA sequencing. These experiments were performed under standard CRISPR-Cas9 genome-editing conditions, as in *Ran et al., 2013*. (B) Flow cytometry analysis reveals an increase in GFP-positive cells with chromatin relative to DNA donor templates. HDR experiments were performed, as outlined in A with MCF10A cells and *GAPDH*, *RAB11A*, or *ACTB* donor templates. The population of GFP-positive cells was gated based on control cells that show no GFP expression (no donor template; upper panel; see also *Figure 1—figure supplement 3*). Representative data from one out of three independent experiments are shown. The results of the other two biological replicates are in *Figure 1—figure supplement 4*. The percentage of GFP-positive cells is indicated in each plot. FSC-A: forward scatter area. (C) Individual results from three independent experiments with each of the target loci. The data points from each independent experiment are designated with the same colored dots. The mean and standard deviation are indicated for each set of experiments. The *p*-values were determined by using Welch's t test. **, *p* <0.01; *, *p* <0.05. The calculated *p*-values are as follows: *p* = 0.0062 for the *GAPDH* data set; *p* = 0.017 for the *RAB11A* data set; *p* = 0.048 for the *ACTB* data set. (D) The use of chromatin relative to naked DNA donor templates results in a 2.3- to 7.4-fold enhancement of GFP-positive cells. The data for each of three independent HDR experiments with each locus are shown. The bars represent mean and standard deviation for each locus.

The online version of this article includes the following figure supplement(s) for figure 1:

**Figure supplement 1.** Schematic representations of the CRISPR-Cas9 target regions for HDR-mediated insertion of a GFP reporter sequence.
**Figure supplement 2.** Reconstitution of plasmid DNA donor templates into chromatin.
**Figure supplement 3.** Flow cytometry analysis of MCF10A cells in control experimental conditions.
**Figure supplement 4.** Flow cytometry analyses of biological replicates of HDR-mediated gene integration experiments in MCF10A cells.

---

chromatin relative to naked DNA donor templates can increase the efficiency of homozygous editing.

We also observed imperfect editing, in which there was at least one improperly edited chromosome, as indicated by either the absence of an edited chromosome or the presence of a PCR product whose size is not consistent with that of an edited or wild-type chromosome. In addition, by performing long-range PCR as in *Kosicki et al., 2018*, we identified two apparently homozygous clones that contained one chromosome with a precisely edited allele and one chromosome with a large deletion at the other allele (*Figure 2—figure supplement 2*). Hence, in the generation of homozygous clones, it is important to carry out both standard and long-range PCR analyses.

The overall efficiency of achieving homozygous editing in diploid MCF10A cells was 15-fold (7.4 $\times$ 2.1) at the *GAPDH* locus, 3.2-fold (2.9 $\times$ 1.1) at the *RAB11A* locus, and large but not quantifiable at the *ACTB* locus, at which we saw homozygous editing only with a chromatin donor template. The *ACTB* locus serves as an example in which the use of a chromatin template relative to a naked DNA template was the difference between a successful and an unsuccessful HDR experiment.

To determine whether a chromatin donor template affects the efficiency of HDR in a different cell line, we examined the insertion of GFP sequences at the *GAPDH* locus in HeLa cells, which are human cervical adenocarcinoma cells that are widely used in biomedical research. HeLa cells are aneuploid and contain four copies of the *GAPDH* gene, which is located on chromosome 12. In these experiments, we observed that the use of a chromatin donor template results in a 2.3-fold increase (average of three biological replicates) in the efficiency of insertion of the GFP sequence in at least one *GAPDH* locus in HeLa cells (*Figure 3A, B* and *Figure 3—figure supplement 1*). We then examined the formation of homozygous edited clones that are generated upon targeted insertion of the GFP sequence at all four copies of the *GAPDH* locus in HeLa cells. In this analysis, we found a substantial increase (5/18 clones versus 1/21 clones) in the efficiency of formation of homozygous clones with the use of a chromatin donor template instead of a naked DNA donor template (*Figure 3C,D*,

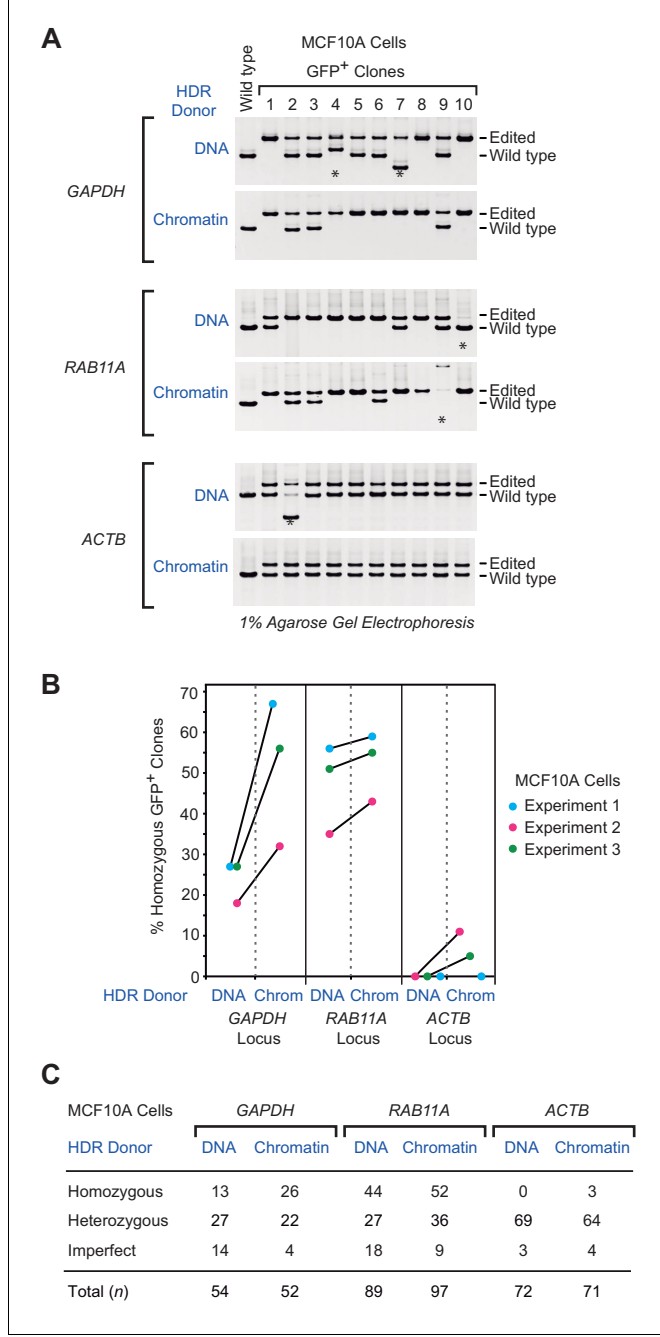

**Figure 2.** The use of chromatin donor templates increases the efficiency of HDR-mediated homozygous gene editing relative to that seen with DNA donor templates. (**A**) PCR analysis of gDNA from MCF10A GFP-positive clones. Three independent HDR experiments were performed as shown in *Figure 1A*, and the gDNA from individual GFP-positive clones was analyzed by PCR. The positions of the PCR amplification products from edited and wild-type alleles are indicated. The PCR products derived from control wild-type cells are also included (left lane of each panel). The asterisks indicate imperfect clones that appear to contain at least one improperly edited chromosome, as indicated by either the absence of an edited chromosome or the presence of a PCR product whose size is not consistent with that of an edited or wild-type chromosome. The positions of the primer pairs (F1, R1) in the PCR analysis of each locus are shown in *Figure 2—figure supplement 1*. The results from a representative subset of the GFP-positive clones are shown. The complete set of PCR results are in *Figure 2—figure supplements 2*, *3* and *5*. (**B**) The percentages of GFP-positive homozygous clones in three independent HDR experiments at each of the target loci. The results from each independent experiment (with DNA versus chromatin donor templates) are denoted with a connector line. The *p*-values were determined by using Welch's

*Figure 2 continued on next page*

*Figure 2 continued*

t-test. The calculated *p*-values are as follows: *p* = 0.062, *p* = 0.56, and *p* = 0.17 for the *GAPDH*, *RAB11A* and *ACTB* data sets, respectively. (C) Summary of the PCR analysis. MCF10A cells are diploid, and each clone was classified as homozygous (with two precisely edited chromosomes), heterozygous (with one precisely edited chromosome and one wild-type chromosome), or imperfect, as defined in A.

The online version of this article includes the following figure supplement(s) for figure 2:

**Figure supplement 1.** Diagrams of the positions of the primer sets for the PCR analysis of GFP-positive clones at the *GAPDH*, *RAB11A*, and *ACTB* loci.
**Figure supplement 2.** PCR analysis of gDNA from GFP-positive clones at the *GAPDH* locus in MCF10A cells.
**Figure supplement 3.** PCR analysis of gDNA from GFP-positive clones at the *RAB11A* locus in MCF10A cells.
**Figure supplement 4.** Long-range PCR analysis of gDNA from GFP-positive clones at the *RAB11A* locus in MCF10A cells.
**Figure supplement 5.** PCR analysis of gDNA from GFP-positive clones at the *ACTB* locus in MCF10A cells.

---

*E* and *Figure 3—figure supplement 2*). Hence, these results show a strong enhancement of HDR by using a chromatin relative to a naked DNA donor template in HeLa cells.

We additionally tested the effect of varying the amount of donor template DNA (as chromatin or naked DNA) upon the efficiency of HDR (*Figure 3—figure supplement 3*). To this end, we used 0.5, 1.0, and 1.5 times the mass of DNA as in a standard experiment with the *GAPDH* donor template in HeLa cells. At each of the three amounts of donor template, we consistently saw a higher efficiency of generation of GFP-positive cells with chromatin relative to naked DNA. Moreover, there was an increase in the fold-enhancement by chromatin as the amount of donor template was increased. We thus observed that a chromatin donor template functions better than a naked DNA donor template for HDR at different concentrations.

Because chromatin has rarely been used in cell transfection experiments, we also investigated the toxicity of chromatin relative to naked DNA in five different human cell lines (*Figure 3—figure supplement 4*). These experiments revealed that chromatin is of comparable or lower toxicity to cells relative to naked DNA in transfection experiments. This low toxicity of chromatin to cells could be useful for HDR applications in which there is low cell viability after transfection.

## Discussion

Here we show that the efficiency of HDR-mediated gene editing can be increased by using a chromatin donor template instead of a naked DNA donor template. Why is chromatin more effective as an HDR donor template than naked DNA? We suggest that chromatin, as the natural form of DNA in the eukaryotic nucleus, is the preferred substrate (relative to naked DNA) for the factors that mediate homologous recombination in cells. In previous biochemical studies, we and others found that eukaryotic Rad51 and Rad54, but not bacterial RecA, can mediate homologous strand pairing, an early step in HDR, with a chromatin donor template (*Alexiadis and Kadonaga, 2002*; *Jaskelioff et al., 2003*). Moreover, we observed that homologous strand pairing occurs more efficiently with a chromatin donor template than with a naked DNA donor template (*Alexiadis and Kadonaga, 2002*). Hence, the new findings on HDR with chromatin donor templates in cells are consistent with the results of the earlier biochemical studies on homologous strand exchange.

In general, a wide range of efficiencies of HDR has been observed in different cell types and with different methodologies. A common factor in these HDR experiments has been, however, the use of a non-chromatin donor template. In this work, we sought to focus specifically on directly comparing the relative efficiencies of HDR with chromatin versus naked DNA donor templates. In these experiments, we consistently observed a higher efficiency of HDR with chromatin relative to naked DNA. These effects include the increased efficiency of targeted insertion of GFP sequences in both loci of a diploid chromosome and in all loci of a tetraploid chromosome. These findings therefore suggest that the use of a chromatin donor template instead of a naked DNA donor template would be a broadly useful strategy for the precise insertion or replacement of DNA sequences via HDR with different methods. Moreover, transfection of chromatin donor templates, which can be simply prepared by salt dialysis methodology with purified DNA and core histones, does not affect cell

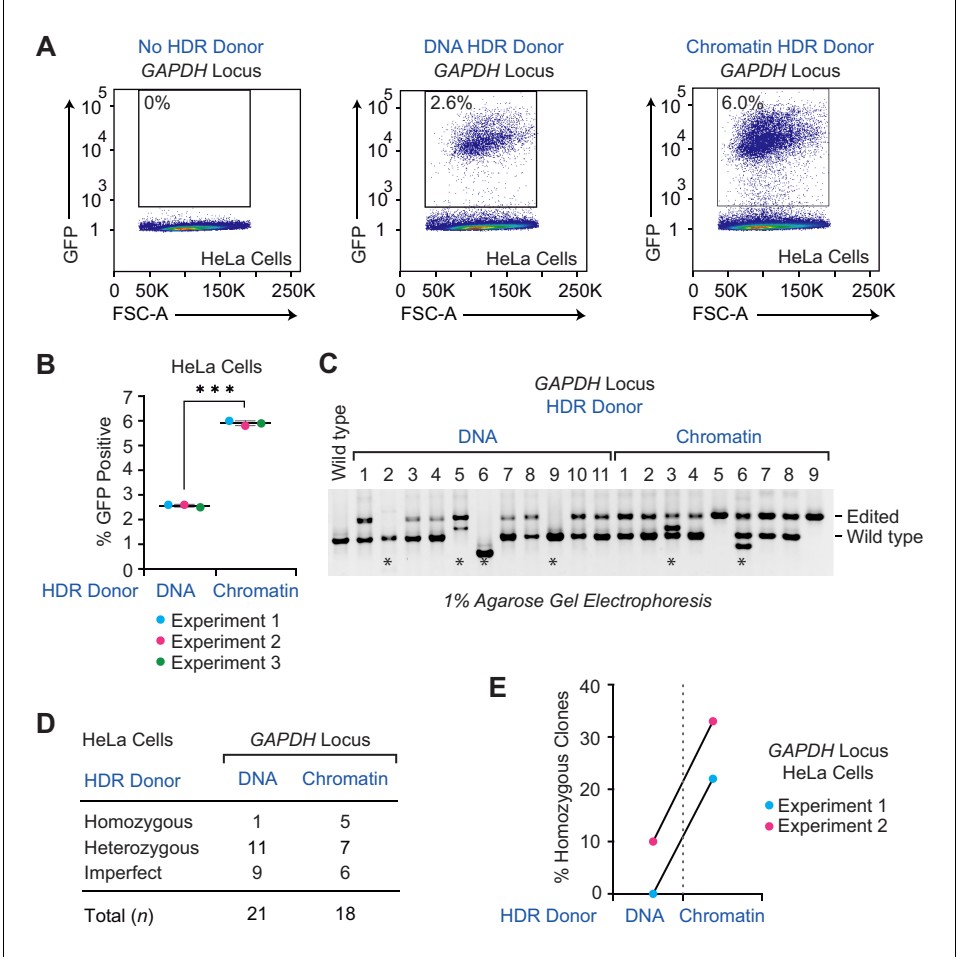

**Figure 3.** The efficiency of HDR-mediated gene editing with CRISPR-Cas9 is higher with a chromatin donor template than with a DNA donor template in HeLa cells. (**A**) The use of a chromatin donor template relative to a naked DNA donor template results in an increase of GFP-positive cells. HDR experiments were performed as depicted in *Figure 1A* with HeLa cells and the *GAPDH* locus donor template. The population of GFP-positive cells was gated based on control cells that show no GFP expression (no HDR donor; left panel). Representative data from one out of three independent experiments are shown. The results of the other two biological replicates are in *Figure 3—figure supplement 1*. The percentage of GFP-positive cells is indicated in each plot. FSC-A: forward scatter area. (**B**) Individual results of flow cytometry analysis from three independent experiments with the *GAPDH* locus and HeLa cells. The data points from each independent experiment are designated with the same colored dots. The *p*-value was determined by using Welch's t-test. ***, *p* <0.0001. The mean and standard deviation are indicated. (**C**) The use of a chromatin HDR donor template results in an increase in the efficiency of homozygous edited clones relative to that seen with a DNA donor template. PCR analysis of edited genomic DNA was carried out as in *Figure 2A*. The positions of the PCR amplification products from edited and wild-type chromosomes are shown. The PCR products from control wild-type cells are also included (left lane). The results from a representative subset of the GFP-positive clones are shown. The results from the other GFP-positive clones that were analyzed are in *Figure 3—figure supplement 2*. (**D**) Summary of the PCR analysis of clones obtained in the HDR-mediated insertion of GFP sequences at the *GAPDH* locus in HeLa cells. The homozygous clones have four copies of the integrated GFP sequence, the heterozygous clones have one to three copies of the integrated GFP sequence, and the imperfect clones appear to contain improperly edited chromosomes, as indicated by either the absence of an edited chromosome or the presence of a PCR product whose size is not consistent with that of an edited or wild-type chromosome. (**E**) The percentages of GFP-positive homozygous clones in two independent HDR experiments. The results from each independent experiment (with DNA versus chromatin donor templates) are denoted with a connector line.

The online version of this article includes the following figure supplement(s) for figure 3:

**Figure supplement 1.** Flow cytometry analyses of biological replicates of HDR-mediated gene integration experiments in HeLa cells.

*Figure 3 continued on next page*

Figure 3 continued

**Figure supplement 2.** PCR analysis of gDNA from GFP-positive clones in HeLa cells.

**Figure supplement 3.** The efficiency of GFP insertion with different amounts of donor template in HeLa cells is higher with chromatin than with DNA.

**Figure supplement 4.** Chromatin templates are of comparable or lower toxicity to cells relative to naked DNA templates.

---

viability. Thus, current methods for HDR can be easily adapted to include chromatin donor templates in place of their naked DNA counterparts.

In this regard, it is notable that we reconstituted chromatin by using native core histones from *Drosophila* embryos. These histones contain an undefined broad mixture of covalent histone modifications (*Levenstein and Kadonaga, 2002*). Because the core histones and their modifications are highly conserved throughout eukaryotes, it seems likely that similar results would be obtained with core histones from other sources. It is possible, however, that the magnitude of enhancement of HDR by chromatin could be further increased by variation of the core histone sequences and modifications.

In conclusion, although there are excellent techniques for the alteration of short (<50 bp) stretches of DNA (*Rees and Liu, 2018*; *Molla and Yang, 2019*; *Anzalone et al., 2019*), there remains a need for increasing the efficiency of the specific insertion or replacement of longer DNA segments that may contain sequences such as the coding regions of genes. We anticipate that chromatin donor templates might be particularly useful for such applications. In addition, we expect that many new gene editing techniques will be developed in the future, and that some of these methods will benefit from the use of chromatin donor templates. Furthermore, the low toxicity of chromatin to cells may be useful for many current and future methods. There is considerable potential to the use of the natural form of the donor template in gene editing experiments. It is our hope that these findings will advance the utility of precise genome editing in basic, translational, and clinical research.

## Materials and methods

To ensure the reproducibility of the results, at least two biological replicates were performed for each experimental condition. The exact number of replicates of each experiment is indicated in its associated figure legend.

### DNA constructs

CRISPR RNA (crRNA) sequences targeting the *GAPDH*, *RAB11A*, or *ACTB* loci were each inserted into the pU6-(BbsI)CBh-Cas9-T2A-mCherry vector (Addgene plasmid # 64324) as described (*Ran et al., 2013*). The crRNA sequences that were used are as follows: *GAPDH*, GAGAGAGACCC TCACTGCTG; *RAB11A*, GGTAGTCGTACTCGTCGTCG; *ACTB*, GGTGAGCTGCGAGAATAGCC. The donor template plasmid for the modification of the *GAPDH* locus was generated as follows. Two homology arm (HA) sequences (~1 kb each) were PCR-amplified with Phusion polymerase (NEB) and genomic DNA (gDNA) from HeLa cells. The oligonucleotides that were used are as follows (the upper case letters are complementary to *GAPDH* or T2A-EGFP sequences): 5' HA, agagataagcttG-GACACGCTCCCCTGACTT, agagatggatccCTCCTTGGAGGCCATGTGGG; 3' HA, tgatagggtaccCC TGCCACACTCAGTCCC, tgataggaattcGCTGGGGGTTACAGGCGTGCG. The T2A-EGFP sequence was PCR-amplified from the PX461 plasmid (Addgene plasmid # 48140) with the following oligonucleotides: agagatggatccGAGGGCAGAGGAAGTCTGCT and agagatggtaccTTACTTGTACAGCTCG TCCA. Then, the three DNA fragments were sequentially subcloned into the pBluescript KS vector (Stratagene). The 3' HA sequence was inserted between the KpnI and EcoRI sites; the T2A-EGFP sequence was inserted between the BamHI and the KpnI sites; and the 5' HA sequence was inserted between the HindIII and the BamHI sites. All restriction enzymes were from NEB. The donor template plasmid for the modification of the *RAB11A* locus was Addgene plasmid # 112012, and the donor template plasmid for the modification of the *ACTB* locus was Addgene plasmid # 87425.

## Chromatin reconstitution

Native *Drosophila* core histones from embryos collected from 0 to 12 hr after egg deposition were purified as described (*Fyodorov and Levenstein, 2002*; *Khuong et al., 2017*). The donor repair template plasmids were purified with the HiSpeed plasmid kit (Qiagen). The optimal histone:DNA ratio for each donor repair template was determined by carrying out a series of reactions with different histone:DNA ratios and then assessing the quality of chromatin by the micrococcal nuclease digestion assay, as described (*Fyodorov and Levenstein, 2002*; *Khuong et al., 2017*). Chromatin was reconstituted with purified core histones by using the salt dialysis method (*Stein, 1989*; *Fei et al., 2015*). In a typical chromatin reconstitution reaction, 50 µg plasmid DNA and 50 µg core histones were combined in TE buffer (10 mM Tris-HCl, pH 8, containing 1 mM EDTA) containing 1 M NaCl in a total volume of 150 µL. The mixture was dialyzed at room temperature against the following buffers in the indicated order: 2 hr in TE containing 0.8 M NaCl; 3 hr in TE containing 0.6 M NaCl; 2.5 hr in TE containing 50 mM NaCl. The quality of the resulting chromatin was assessed by using the micrococcal nuclease digestion assay, and the chromatin was stored at 4°C until use.

## Cell lines

HeLa cells were a gift from Dr. Anjana Rao (La Jolla Institute for Immunology). MCF10A cells were a gift from Dr. Jichao Chen (The University of Texas MD Anderson Cancer Center). The MCF10A and HeLa cells were not authenticated. The MCF10A cells and HeLa cells were tested for mycoplasma and found to be negative for mycoplasma contamination.

## Cell culture

MCF10A cells (non-tumorigenic mammary epithelial cells) were maintained in DMEM/F-12 medium (Gibco) supplemented with 20 ng/mL EGF, 500 ng/mL hydrocortisone (Sigma), 10 µg/mL insulin (Sigma), 100 ng/mL cholera toxin (Sigma), 100 U/mL penicillin and 100 µ/mL streptomycin (Gibco), and 5% horse serum (Gibco) at 37°C and 5% $CO_2$. HeLa cells (human cervical carcinoma cells), HT1080 cells (human fibrosarcoma cells), SW480 cells (human colorectal adenocarcinoma cells), and 293 T cells (derived from primary human embryonic kidney cells) were maintained in DMEM, high glucose medium (Corning) supplemented with 10% fetal bovine serum (Gibco) and 100 U/mL penicillin and 100 µ/mL streptomycin (Gibco) at 37°C and 5% $CO_2$.

## Cell transfection

In each series of experiments, cell transfections with chromatin or DNA donor templates were performed by following standard protocols under exactly the same conditions. Transfection of HeLa cells was performed with Lipofectamine 3000 (Invitrogen) according to the manufacturer's recommendations. Linear polyethylenimine (PEI 25K; 25,000 MW; Polysciences, Inc) was used for transfection of MCF10A cells at a PEI:DNA mass ratio of 3:1. The transfections were performed as follows. 5 $\times 10^5$ cells/well were plated in six well plates the day before transfection. For each CRISPR-Cas9 target locus, cells were co-transfected with equal amounts of the target-specific donor repair template (as free plasmid DNA or chromatin) and the Cas9 coding plasmid containing the target-specific single guide RNA sequence. For HeLa cells, DNA (1.25 µg) or chromatin (containing 1.25 µg of DNA) was used in each transfection (except for the experiment in *Figure 3—figure supplement 3*, in which 1.25 µg of the Cas9 coding plasmid containing the single guide targeting the *GAPDH* locus was co-transfected with 0.625 µg, 1.25 µg, or 1.875 µg of donor template DNA as naked DNA or chromatin); for MCF10A cells, DNA (1.5 µg) or chromatin (containing 1.5 µg of DNA) was used in each transfection.

## FACS and flow cytometry analysis

At 24 hr post-transfection, cells were detached with 0.25% trypsin (Corning). After centrifugation, the cell pellets were resuspended in culture media containing 250 ng/mL DAPI (Sigma). mCherry-positive, DAPI-negative cells were sorted by FACS and collected in six well plates (HeLa cells; 100,000 cells/well) or 24 well plates (MCF10A cells; 30,000 cells/well). Then, the cells were passaged twice before the analysis of the expression of GFP by flow cytometry. GFP-positive single-cells were sorted by FACS into 96 well plates. To determine the percentage of GFP-positive cells, at least 100,000 cells of each condition were analyzed by flow cytometry with a BD FACSAria Fusion or a BD

FACSAria2 instrument at the Human Embryonic Stem Cell Core Facility (UCSD). The BD FACSDiva Software was used for data acquisition, and data analysis was performed with FlowJo version 10.6.1 (BD).

## Molecular analysis of the targeted loci

Genomic DNA samples from wild-type cells as well as from independent GFP-positive clones were isolated with the Quick Extract DNA extraction solution (Lucigen) by following the manufacturer's recommendations, and were then subjected to PCR analysis. First, the occurrence of edited alleles was analyzed with primers that flank the 5' and 3' homology arm sequences (and thus do not contain sequences in the donor template) at the location in which the GFP DNA was inserted. The specific primers that were used are as follows: *GAPDH*, F1: TGACAACAGCCTCAAGATCATCAGG, R1: GA TGGAGTCTCATACTCTGTTGCCT; *RAB11A*, F1: TGGGAAGTGGACATCATTGG, R1: GACCC TCCAATATGTTCTGT; *ACTB*, F1: AATGCTGCACTGTGCGGCGA, R1: ATGGCATGGGGGAGGGCA TA. Then, genomic DNA from potentially homozygous GFP-positive clones was analyzed by long-range PCR analysis with LongAmp Hot Start *Taq* DNA Polymerase (NEB), as described by *Kosicki et al., 2018*. The primers that were used are as follows. *GAPDH*, F2: CTCCTGCAGTGA TTTGTTTCTTCTT, R2: ACTCATTCTCCCAACACACATCAAA; *RAB11A*, F2: GCTTTATCTTCTTTTTGC TCACCTG, R2: GTGTCCCATATCTGTGCCTTTATTG; *ACTB*, F2: ATGAATAAAAGCTGGAGCACC-CAA, R2: TTGTGCAGCTATACGCAAGATTAAG. The locations of the PCR primers at the *GAPDH*, *RAB11A*, and *ACTB* loci are depicted in *Figure 2—figure supplement 1*. To confirm the integrity of the homozygous clones obtained with chromatin donor templates, we determined the DNA sequences of three *GAPDH* clones and three *ACTB* clones across the insertion junctions and found that the GFP sequences were precisely inserted into the target sites in all six clones.

## Statistical analysis

The two-tailed Welch t-test with alpha = 0.05 was performed by using GraphPad Prism version 8.4.1 (GraphPad Software).

# Acknowledgements

We are grateful to E Peter Geiduschek, George Kassavetis, Jia Fei, Long Vo ngoc, Cassidy Yunjing Huang, Selena Chen, and Claudia Medrano for critical reading of the manuscript. We thank Ralf Kuehn, Feng Zheng, Alexander Marson, and the Allen Institute for Cell Science for the generous gifts of plasmids as well as George Kassavetis for providing bacteriophage T7 DNA. G.CB. is a Pew Latin American Postdoctoral Fellow. JT.K is the Amylin Chair in the Life Sciences. This work was supported by a grant from the National Institutes of Health (R35 GM118060) to J.TK.

# Additional information

## Competing interests

Grisel Cruz-Becerra, James T Kadonaga: have filed a patent application (PCT/US2019/029194) that describes the invention reported in this article.

## Funding

| Funder | Grant reference number | Author |
| --- | --- | --- |
| National Institutes of Health | R35 GM118060 | James T Kadonaga |

The funders had no role in study design, data collection and interpretation, or the decision to submit the work for publication.

## Author contributions

Grisel Cruz-Becerra, Conceptualization, Formal analysis, Validation, Investigation, Methodology, Writing - original draft, Writing - review and editing; James T Kadonaga, Conceptualization, Formal analysis, Supervision, Funding acquisition, Writing - original draft, Writing - review and editing

## Author ORCIDs
Grisel Cruz-Becerra (iD) https://orcid.org/0000-0001-6297-4132
James T Kadonaga (iD) https://orcid.org/0000-0002-2075-9458

## Decision letter and Author response
Decision letter https://doi.org/10.7554/eLife.55780.sa1
Author response https://doi.org/10.7554/eLife.55780.sa2

## Additional files

### Supplementary files
• Transparent reporting form

### Data availability
All data generated or analysed during this study are included in the manuscript and supporting files.

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
