## [Decision Letter]

**Acceptance summary:**

The targeted integration of DNA sequences constitutes an important aspect of gene editing. Current techniques are quite inefficient, and the authors found that chromatinized templates enhance targeting efficiencies at three different gene loci. In addition, the authors report an increase in targeting both alleles using chromatinized templates. These observations constitute a critical technical advance that will facilitate certain gene targeting experiments.

**Decision letter after peer review:**

Thank you for submitting your article "Enhancement of homology-directed repair with chromatin donor templates in cells" for consideration by *eLife*. Your article has been reviewed by three peer reviewers, including Wolf-Dietrich Heyer as the Reviewing Editor and Reviewer #1, and the evaluation has been overseen by a Reviewing Editor and Jessica Tyler as the Senior Editor. The following individuals involved in review of your submission have agreed to reveal their identity: Michael Lieber (Reviewer #2); Jeremy Stark (Reviewer #3).

The reviewers have discussed the reviews with one another and the Reviewing Editor has drafted this decision to help you prepare a revised submission.

Summary:

The targeted integration of DNA sequences constitutes an important aspect of gene editing. Current techniques are quite inefficient, and the authors found that chromatinized templates enhance targeting efficiencies at three different gene loci. In addition, the authors report an increase in bi-allelic targeting events using chromatinized templates. These observations constitute a critical technical advance that will facilitate certain gene targeting experiments.

The experiment involves co-transfection of the chromatin donor, along with plasmids expressing Cas9 and single guide RNAs. A few different targeted loci are tested in MCF10A cells, which is an appropriate model cell line, and one is tested in HeLa cells. The donor has a promoterless GFP, so homologous integration should cause GFP+ cells. As well, through isolation of GFP+ clones, targeted integration is confirmed, and an apparent increase in homozygous targeting is shown. Altogether, the approaches are appropriate to test the hypothesis that a chromatin donor causes a greater frequency of homologous recombination, compared to a protein-free donor. These findings will likely be of broad interest, however, there are issues of rigor and controls that should be addressed.

Essential revisions:

1) There is a concern with the limited evidence of reproducibility. Throughout the manuscript, it appears that representative n=3 experiments are shown, without statistics. In the Materials and methods section, it is stated that 2 biological replicates were performed, but the data do not appear to be shown. All the frequency data for experiments that had working controls should be shown. The raw HR frequencies will vary between experiments, which is fine, but such data should be shown. In particular, in each figure, for the flow cytometry frequency data, data from additional replicates should be shown, and in the main figure. This will give the reader the sense of variability among experiments for the raw data, as well as any variability in the fold-effect for the chromatin DNA template at individual targeted sites.

2) There are no statistics in the manuscript. In addition to statistics for the effect of the chromatin template for frequencies of GFP+ cells, for the clonal analysis, an appropriate statistical test should be employed for homozygous vs. heterozygous targeting.

3) The nature of the chromatin should be spelled out at the beginning of the Results section. What is known about the nature of the native *Drosophila* chromatin used? Does it tend to have a broad mix of histone tail modifications? How do these histone tail modifications relate to what is found in human cell lines? If the precise nature of the chromatin is not well understood, at least this should be stated in the Results section and Discussion section.

4) An unlikely issue with the PCR confirmation is the possibility of primer jumping that will artificially appear as targeted clones. The homozygous targeted clones could also reflect one targeted allele, and a large deletion on the other allele, based on the work of Allan Bradley (PMID: 30010673). While these possibilities may be unlikely, some PCR analysis of homozygous clones using more distant primers, again as performed by the Bradley laboratory (e.g. with LongAMP polymerase / NEB) would help ensure that these are bona fide targeted events.

---

## [Author Response]

Summary:The targeted integration of DNA sequences constitutes an important aspect of gene editing. Current techniques are quite inefficient, and the authors found that chromatinized templates enhance targeting efficiencies at three different gene loci. In addition, the authors report an increase in bi-allelic targeting events using chromatinized templates. These observations constitute a critical technical advance that will facilitate certain gene targeting experiments.The experiment involves co-transfection of the chromatin donor, along with plasmids expressing Cas9 and single guide RNAs. A few different targeted loci are tested in MCF10A cells, which is an appropriate model cell line, and one is tested in HeLa cells. The donor has a promoterless GFP, so homologous integration should cause GFP+ cells. As well, through isolation of GFP+ clones, targeted integration is confirmed, and an apparent increase in homozygous targeting is shown. Altogether, the approaches are appropriate to test the hypothesis that a chromatin donor causes a greater frequency of homologous recombination, compared to a protein-free donor. These findings will likely be of broad interest, however, there are issues of rigor and controls that should be addressed.

We thank Wolf, Michael, Jeremy, and Jessica for their positive and enthusiastic assessment of this work and for their insightful recommendations. We have incorporated all of the recommended modifications into the revised manuscript as follows.

1) We added the requested raw data.

2) We added statistical analyses, and display the data as scatter plots instead of bar graphs.

3) We added a discussion of the core histones.

4) We performed the long-range PCR with LongAmp polymerase.

I might also mention that the revised text also uses the term "homozygous" instead of "biallelic" to describe the precise integration into both alleles of the target loci in the diploid MCF10A cells.

We appreciate your very constructive and helpful recommendations that have substantially improved the quality of this work.

Essential revisions:1) There is a concern with the limited evidence of reproducibility. Throughout the manuscript, it appears that representative n=3 experiments are shown, without statistics. In the Materials and methods section, it is stated that 2 biological replicates were performed, but the data do not appear to be shown. All the frequency data for experiments that had working controls should be shown. The raw HR frequencies will vary between experiments, which is fine, but such data should be shown. In particular, in each figure, for the flow cytometry frequency data, data from additional replicates should be shown, and in the main figure. This will give the reader the sense of variability among experiments for the raw data, as well as any variability in the fold-effect for the chromatin DNA template at individual targeted sites.

Thank you for this helpful request and clarification, as we were not quite sure about exactly what to include in the manuscript. We now clearly indicate in the revised manuscript that we performed three biological replicates for the HDR experiments at each of the three target loci in MCF10A cells and three biological replicates at the *GAPDH* locus in HeLa cells.

We have now included all of the raw data for these experiments as follows.

a) Flow cytometry. We added the following raw data:

New Figure 1—figure supplement 4

New Figure 3—figure supplement 1

New Figure 3—figure supplements 3A and 3B

b) Standard PCR experiments. We added the following raw data:

New Figure 2—figure supplement 2A and 2B

New Figure 2—figure supplement 3A and 3B

New Figure 2—figure supplement 5A and 5B

New Figure 3—figure supplement 2A

c) New long-range PCR experiments (point 4, below). We added the following raw data:

New Figure 2—figure supplement 2C

New Figure 2—figure supplement 4A and 4B

New Figure 2—figure supplement 5D

New Figure 3—figure supplement 2B

d) In addition, we have now revised the bar graphs to scatter plots. The scatter plots show the individual data points rather than the averages, and are as follows:

Revised Figure 1C

Revised Figure 1D

Revised Figure 2B

Revised Figure 3B

Revised Figure 3E

Revised Figure 3—figure supplement 3C

2) There are no statistics in the manuscript. In addition to statistics for the effect of the chromatin template for frequencies of GFP+ cells, for the clonal analysis, an appropriate statistical test should be employed for homozygous vs. heterozygous targeting.

Thank you very much for this recommendation, which enhances the quality of the analysis. We analyzed the data by using Welch's t-test because it is more robust than the more commonly used Student's t-test (with which we obtained lower *p*-values than with Welch's t-test). The *p-*values for the experiments are now given in the revised figure legends. In addition, as mentioned above in point 1 (part d), we have replaced the bar graphs with scatter plots. The scatter plots are much more informative than the bar graphs.

3) The nature of the chromatin should be spelled out at the beginning of the Results section. What is known about the nature of the native *Drosophila* chromatin used? Does it tend to have a broad mix of histone tail modifications? How do these histone tail modifications relate to what is found in human cell lines? If the precise nature of the chromatin is not well understood, at least this should be stated in the Results section and Discussion section.

Thank you for this question. We previously analyzed the core histones from *Drosophila* embryos by mass spectrometry (Levenstein and Kadonaga, 2002), and found that the histones contain a broad mixture of covalent modifications. The precise nature of covalent modifications has not been resolved.

As recommended, we added the following sentence at the beginning of the Results section.

“The chromatin was reconstituted by using salt dialysis methodology with plasmid DNA and purified core histones from *Drosophila* embryos, which contain a broad mixture of covalent modifications that have not been precisely resolved (Levenstein and Kadonaga, 2002).”

We also added the following paragraph in the Discussion section.

“In this regard, it is notable that we reconstituted chromatin by using native core histones from *Drosophila* embryos. These histones contain an undefined broad mixture of covalent histone modifications (Levenstein and Kadonaga, 2002). Because the core histones and their modifications are highly conserved throughout eukaryotes, it seems likely that similar results would be obtained with core histones from other sources. It is possible, however, that the magnitude of enhancement of HDR by chromatin could be further increased by variation of the core histone sequences and modifications.”

4) An unlikely issue with the PCR confirmation is the possibility of primer jumping that will artificially appear as targeted clones. The homozygous targeted clones could also reflect one targeted allele, and a large deletion on the other allele, based on the work of Allan Bradley (PMID: 30010673). While these possibilities may be unlikely, some PCR analysis of homozygous clones using more distant primers, again as performed by the Bradley laboratory (e.g. with LongAMP polymerase / NEB) would help ensure that these are bona fide targeted events.

Thank you very much for this helpful suggestion. We performed long-range PCR on 115 of our apparently homozygous clones with LongAmp Hot Start *Taq* DNA Polymerase (NEB), as in Kosicki et al., (2018) (new Figure 2—figure supplement 2, Figure 2—figure supplement 4, and Figure 2—figure supplement 5; new Figure 3—figure supplement 2). We found that two out of the 115 apparently homozygous clones appeared to contain one chromosome with a properly edited allele and one chromosome with a large deletion in the other allele (new Figure 2—figure supplement 2). Hence, as recommended by the reviewers, these results thus highlight the importance of carrying out long-range PCR analysis of apparently homozygous clones. In addition to the new figures cited above, the text has been modified as follows.

“In addition, by performing long-range PCR as in Kosicki et al., (2018), we identified two apparently homozygous clones that contained one chromosome with a precisely edited allele and one chromosome with a large deletion at the other allele (Figure 2—figure supplement 2). Hence, in the generation of homozygous clones, it is important to carry out both standard and long-range PCR analyses.”